# Genome-Wide Characterization of the PIFs Family in Sweet Potato and Functional Identification of *IbPIF3.1* under Drought and *Fusarium* Wilt Stresses

**DOI:** 10.3390/ijms24044092

**Published:** 2023-02-17

**Authors:** Nan Nie, Jinxi Huo, Sifan Sun, Zhidan Zuo, Yanqi Chen, Qingchang Liu, Shaozhen He, Shaopei Gao, Huan Zhang, Ning Zhao, Hong Zhai

**Affiliations:** 1Key Laboratory of Sweet Potato Biology and Biotechnology, Ministry of Agriculture and Rural Affairs/Beijing Key Laboratory of Crop Genetic Improvement/Laboratory of Crop Heterosis and Utilization, Ministry of Education, College of Agronomy & Biotechnology, China Agricultural University, Beijing 100193, China; 2Institute of Sericulture and Tea, Zhejiang Academy of Agricultural Sciences, Hangzhou 310021, China

**Keywords:** *Ipomoea batatas*, *Ipomoea triloba*, *Ipomoea trifida*, PIFs, expression analysis, function analysis

## Abstract

Phytochrome-interacting factors (PIFs) are essential for plant growth, development, and defense responses. However, research on the *PIFs* in sweet potato has been insufficient to date. In this study, we identified *PIF* genes in the cultivated hexaploid sweet potato (*Ipomoea batatas*) and its two wild relatives, *Ipomoea triloba*, and *Ipomoea trifida*. Phylogenetic analysis revealed that IbPIFs could be divided into four groups, showing the closest relationship with tomato and potato. Subsequently, the PIFs protein properties, chromosome location, gene structure, and protein interaction network were systematically analyzed. RNA-Seq and qRT-PCR analyses showed that *IbPIFs* were mainly expressed in stem, as well as had different gene expression patterns in response to various stresses. Among them, the expression of *IbPIF3.1* was strongly induced by salt, drought, H_2_O_2_, cold, heat, *Fusarium oxysporum* f. sp. *batatas* (*Fob*), and stem nematodes, indicating that *IbPIF3.1* might play an important role in response to abiotic and biotic stresses in sweet potato. Further research revealed that overexpression of *IbPIF3*.1 significantly enhanced drought and *Fusarium* wilt tolerance in transgenic tobacco plants. This study provides new insights for understanding PIF-mediated stress responses and lays a foundation for future investigation of sweet potato PIFs.

## 1. Introduction

Light not only provides photosynthates and energy to plants, but is also an important environmental stimulus regulating plant growth and defense [1]. Plants respond to abiotic and biotic stresses by sensing changes in light wavelength, intensity, direction, and duration [2,3]. Photoreceptors play an essential role in light signal reception. Several photoreceptors have been found in plants, including phytochrome (PHY), cryptochrome (CRY), UV-B photoreceptor (UVR8), and phototropin [4,5,6], which are involved in a series of physiological and biochemical reactions in plants, such as photomorphogenesis, abiotic stress tolerance, and plant defense [7,8,9,10]. Phytochrome-interacting factors (PIFs) interact physically with the red and far-red light photoreceptors to mediate light responses [11,12,13]. When exposed to light, PHY promotes the rapid phosphorylation, sequential ubiquitination, and eventual degradation of PIFs [14]. Research has shown that PIFs can directly regulate the expression of downstream genes by binding to G-box (CACGTG) and/or E-box (CANNTG) motifs contained in their promoter [15].

PIFs are a subfamily of basic helix-loop-helix (bHLH) transcription factors [16]. *Arabidopsis* has at least eight PIFs (AtPIF1 to AtPIF8), which either redundantly or exclusively regulate plant growth and development. All AtPIFs contain a bHLH domain, which plays an important role in the formation of AtPIF homodimers and heterodimers [13]. The active phyB-binding (APB) domain is present in all AtPIFs, whereas the active phyA-binding (APA) domain is present only in AtPIF1 and AtPIF3. Aside from *Arabidopsis*, PIFs have been extensively studied in other plants, including 8 found in tomato [17], 4 in grape [18], 8 in apple [19], 7 in tea [20], 6 in pepper [21], 14 in peanut [22], 7 in potato [23], 30 in rapeseed [24], and 5 in carrot [25]. 

PIFs have been reported as key regulators of plant growth, development, and metabolism. AtPIF3, AtPIF4, and AtPIF7 promote *Arabidopsis* hypocotyl elongation [26]. RhPIF8 regulates rose petal senescence by modulating ROS homeostasis [27]. The overexpression of *OsPIL1/OsPIL13* in transgenic rice plants promotes internode elongation [28]. *SlPIF3* is involved in tocopherol biosynthesis during tomato fruit ripening [29]. In addition, accumulating evidence has indicated that PIFs are essential factors regulating the responses to various abiotic stresses. *ZmPIF1* and *ZmPIF3* have been shown to enhance drought tolerance in rice [30,31]. AtPIF3, AtPIF4, and AtPIF7 regulate plant cold tolerance by repressing *CBF/DREB1* gene expression [32,33,34]. In *Arabidopsis*, AtPIF4 inhibits secondary cell wall thickening and induces the shade avoidance [35]. The role of PIFs in regulating plant defense responses has also been studied. AtPIF4 acts as a negative regulator of immunity and increases susceptibility to *Pseudomonas syringae* pv. tomato DC3000 [36]. PIF1/3/4/5 negatively regulate *Arabidopsis* resistance to *Botrytis cinerea* during plant defense against necrotic pathogens [37]. In recent years, increasingly more studies have noted that PIFs play an important role in increasing crop yields. Knockout of *OsPIL15* using CRISPR/Cas9 improves grain size and weight in rice [38]. *ZmPIF1* enhances rice yield by increasing tiller number and panicle number [30]. However, the related molecular mechanism in sweet potato is still poorly understood.

Sweet potato (*Ipomoea batatas* (L.) Lam., 2n = 6x = 90) is a dicotyledonous plant of the Convolvulaceae family. Sweet potato is the seventh largest food crop in the world and has been considered as a new type of bioenergy [39]. Due to its strong adaptability and resistance, it is widely planted in drought, waterlogged, and saline areas [40]. Several genes have been reported to be associated with abiotic stress in sweet potato, including *IbMIPS1*, *IbC3H18*, *IbBBX24*, *ItfWRKY70*, *IbPYL8*, *IbbHLH66*, *IbbHLH118*, and *IbMYB48* [41,42,43,44,45,46]. Sweet potato is susceptible to many pests and diseases, which may cause huge economic losses in crop production. *Fusarium* wilt, a soil-borne pathogenic fungal disease caused by *Fusarium oxysporum* f. sp. *batatas* (*Fob*), is one of the most destructive diseases in sweet potato plants [47]. Once infected with *Fob*, the leaves and veins shrivel and the plant eventually dies [48]. Plant parasitic stem nematodes mainly damage plant roots and underground tissues, seriously affecting the yield and quality of sweet potato [49]. At present, some progress has been made on the defense mechanism of sweet potato. *IbSWEET10* and *IbBBX24* improve resistance to *Fusarium* wilt of transgenic sweet potato [43,50]. *IbMIPS1* significantly increases callose and lignin content and enhances stem nematode resistance in transgenic sweet potato [41]. However, the study of *PIF* genes in sweet potato under stress has not been reported. Due to the complicated and highly heterozygous genetic background of sweet potato, the improvement of its agronomic traits is limited [51]. With the continuous advancement of sequencing technology, the genome assembly of hexaploid sweet potato Taizhong 6 [52], and two diploid species, *Ipomoea triloba* NCNSP0323 (2n = 2x = 30) and *Ipomoea trifida* NCNSP0306 (2n = 2x = 30) have recently been completed [53], making it possible to systematically investigate important gene families in sweet potato. 

In this study, we identify *PIF* genes in *I. batatas*, *I. triloba*, and *I. trifida*. We analyzed the protein properties, chromosomal location, phylogenetic relationship, and structure of the *PIF* genes as well as the *cis*-elements of their promoters and interaction network of the PIF proteins in sweet potato. To further clarify the function of these genes in sweet potato and its two diploid relatives, we tested the expression pattern of *PIFs* in different tissues and under various stress conditions. On this basis, the role of *IbPIF3.1* under drought and *Fob* treatment was preliminarily verified in tobacco.

## 2. Results

### 2.1. Identification and Characteristic of PIFs in Sweet Potato and Its Two Diploid Relatives

To the identified PIFs members, we utilized the protein sequences of PIFs in *Arabidopsis* as queries to perform a BLASTP search against the sweet potato and its two diploid relative protein sequence databases. As a result, a total of 18 protein sequences have been identified in three *Impoea* genomes including 6 of *I. batatas*, 6 of *I. triloba*, and 6 of *I. trifida*. Then, all PIFs members were named according to their homologous genes in *Arabidopsis*. The *PIF* genes from *I. batatas* were named after “*Ib*”; *I. triloba*, named after “*Itb*”; and *I. trifida*, named after “*Itf*” (Appendix A). The basic physicochemical information of IbPIFs were analyzed as listed in Table 1. *IbPIF1.2* and *IbPIF4* had the smallest and largest genomic lengths, respectively, which varied from 3275 bp (*IbPIF1.2*) to 8025 bp (*IbPIF4*). The CDS length of *IbPIFs* ranged from 1365 bp (*IbPIF1.2*) to 2661 bp (*IbPIF4*). The length of the putative proteins ranged from 454 aa to 886 aa while the molecular weight (MW) ranged from 48.728 kDa to 97.031 kDa. The theoretical isoelectric points (pI) of various proteins ranged between 5.29 and 7.17. All of the IbPIF proteins were unstable, with an instability index of more than 43. Their GRAVY scores were less than 0, indicating that they are hydrophilic proteins. Subcellular localization predicted that all of the IbPIFs were located in the nucleus.

### 2.2. Chromosomal Location of Sweet Potato and Its Two Diploid Relatives

Chromosomal location showed that all of the *PIFs* from *I. batatas*, *I. triloba*, and *I. trifida* were dispersed on five chromosomes (Figure 1). In *I. batatas*, one *IbPIF* was on each of Chr9, Chr11, Chr12, and Chr13; and two were on Chr2 (Figure 1A). In *I. triloba* and *I. trifida*, the distribution of *PIFs* was similar: one *PIF* was detected on each of Chr1, Chr2, Chr7, and Chr10; and two were on Chr4 (Figure 1B,C). The results indicated that the distribution of *PIFs* was similar on chromosomes in sweet potato and its two diploid relatives.

### 2.3. Phylogenetic Analysis of PIFs in Sweet Potato and Its Two Diploid Relative

To study the evolutionary relationships of PIF proteins, an unrooted Maximum Likelihood (ML) phylogenetic tree was created based on multiple alignments of the 70 predicted PIF amino acid sequences (i.e., 8 in *Arabidopsis thaliana*, 7 in *Camellia sinensis*, 5 in *Daucus carota*, 6 in *Ipomoea batatas*, 6 in *Ipomoea triloba*, 6 in *Ipomoea trifida*, 7 in *Malus domestica*, 6 in *Oryza sativa*, 8 in *Solanum lycopersicum*, 7 in *Solanum tuberosum*, and 4 in *Vitis vinifera*, Appendix A). Based on the phylogenetic analysis, they were divided into four groups, PIF1 belonged to group I; PIF4 and PIF5 were in group II; PIF2, PIF3, and PIF6 were in group III; and PIF7 and PIF8 were in group IV (Figure 2). Within each group, all IbPIFs were clustered with their corresponding orthologs in *I. triloba* or *I. trifida*. For instance, in group I, IbPIF1.1 was clustered with *I. triloba* ortholog (ItbPIF1.1) and *I. trifida* ortholog (ItfPIF1.1), while IbPIF1.2 was clustered with *I. triloba* ortholog (ItbPIF1.2) and *I. trifida* ortholog (ItfPIF1.2). Moreover, the PIFs from sweet potato had the closest relationship with tomato and potato PIFs.

Furthermore, we predicted 10 motifs in the 70 PIF proteins using the MEME website (Appendix A), where motif 1 and motif 3 contained core sequences of the bHLH and APB domains, respectively. All PIFs had motifs 1 to 4, which seemed to be conserved motifs in PIFs. Most PIFs in the same group had similar conserved motifs and motif distribution, especially in cultivated hexaploid sweet potato and its diploid wild relatives.

### 2.4. Conserved Domain and Exon–Intron Structure Analysis of PIFs in Sweet Potato and Its Two Diploid Relatives

Protein domains and gene structure are important in analyzing and predicting gene functions. The presence of bHLH and APB domains is one of major characteristics of PIF members. Analysis of the 26 PIFs from *A. thaliana*, *I. batatas*, *I. triloba*, and *I. trifida* showed that all PIFs members contained both the bHLH and APB domains (Figure 3A); while IbPIF1.1, PIF3.1, and PIF3.2 contained the APA domain (Figure 3A).

The exon–intron structure of the 26 *PIF* genes were examined, in order to better understand the structural diversity of the *PIFs* (Figure 3B). Most *PIF* genes possessed six to seven exons. *IbPIF4* contained the largest number of exons, at 17. Closely related members of each group usually had similar exon distribution models, with little difference in exon number and length. For example, the *PIFs* of group I had exons ranging from 6 to 9, group II ranged from 6 to 7, and group IV ranged from 6 to 10. In group III, we found a significant difference in the number of *PIF* homologous gene exons among *I. batatas*, *I. triloba*, and *I. trifida*; for example, *ItbPIF4* possessed 7 exons, while *ItfPIF4* possessed 16 exons and *IbPIF4* possessed 17 exons.

### 2.5. cis-Element Analysis in the Promoter of PIFs in Sweet Potato and Its Two Diploid Relatives

PIFs are involved in plant growth, development, and stress response, and *cis*-elements in the promoter region play a key role in expression of *PIF* genes in these processes. In order to further understand the transcriptional regulatory mechanisms of *PIFs*, 2000 bp upstream sequences of *PIFs* from *I. batatas*, *I. triloba*, and *I. trifida* were used to carry out *cis*-element analysis. The results showed that promoters of *PIFs* contained a variety of light-responsive elements including G-box (TACGAT), Box 4 (ATTAAT), ACE (CTAACGTATT), and LAMP (CTTTATCA) [54]; hormonal response elements including abscisic acid (ABA)-responsive element ABRE (ACGTG), gibberellin (GA)-responsive element GARE (TCTGTTG), auxin (AUX)-responsive element TGA (AACGAC), jasmonic acid (JA)-responsive element CGTCA, and salicylic acid (SA) TCA-responsive element (CCATCTTTTT) [55,56]. In addition, various elements responding to abiotic and biotic stresses were also found, such as anaerobic-responsive, drought-responsive, low-temperature-responsive, wound-responsive, and defense- and stress-responsive elements. The *IbPIF3.1*, *IbPIF4,* and *IbPIF8* promoter regions contained a variety of stress-responsive elements, including drought-, and defense- and stress-responsive elements, whereas the *cis*-elements in the *IbPIF1.1*, *IbPIF1.2,* and *IbPIF3.2* promoters were less diverse (Figure 4). Overall, the results indicated that the *PIFs* in sweet potato might participate in an intricate regulatory network to adapt to complicated and changeable environments.

### 2.6. Protein Interaction Network of IbPIFs in Sweet Potato

To investigate the potential regulatory network of IbPIFs, we constructed an IbPIF protein interaction network based on *Arabidopsis* orthologous proteins (Figure 5). Protein interaction prediction indicated that IbPIFs could interact with each other (i.e., PIF1, PIF3, PIF4, PIF5, and PIF7), and also interacted with bHLH family proteins (bHLH119). IbPIFs could interact with a variety of transcription factors to regulate light signaling pathways (i.e., PHYB, PHYA, PAR2, PIA2, FHY3, TOC1, and HFR1). In addition, IbPIFs could interact with other proteins to regulate hormone signaling, such as gibberellin signal transduction components (i.e., RGA, RGL1, RGL2, and RGL3) and brassinosteroid (BR) signaling pathway- related protein BZR1. IbPIF1 and IbPIF3 could interact with a VQ motif-containing protein (VQ29) which regulates various developmental processes and responses to biotic and abiotic stresses. IbPIF4 might interact with HRB1 in response to drought stress. IbPIF1 and IbPIF4 might interact with S-nitrosylation of the small ubiquitin-like modifier (SUMO)-conjugating enzyme 1 (SCE1) to regulate plant immunity. Overall, these results suggested that IbPIFs could participate in multiple regulatory networks interacting with related transcription factors and functional proteins.

### 2.7. Expression Analysis of PIFs in Sweet Potato and Its Two Diploid Relatives

#### 2.7.1. Expression Analysis in Various Tissues

To investigate the potential biological functions of *IbPIFs*, RNA-seq of different tissues including shoot, young leaf, mature leaf, stem, fibrous root, initial tuberous root, expanding tuberous root, and mature tuberous root of Yan252 and Xuzi3 were downloaded. These *PIF* genes were expressed in different tissues, and showed tissue-specific expression. In Yan252 and Xuzi3, compared with the young leaf, *IbPIF3.2*, *IbPIF4*, and *IbPIF8* demonstrated higher expression in mature leaf. *IbPIF1.1*, *IbPIF3.1*, and *IbPIF8* presented a gradually down-regulated trend during root development (Figure 6A,B). In Yan252, *IbPIF1.2* was highly expressed in stem, and *IbPIF1.1* was predominantly expressed in the root (Figure 6A). However, *IbPIF1.1* was primarily expressed in the stem of Xuzi3 (Figure 6B).

To explore the functions of *ItbPIFs* and *ItfPIFs*, their expression profiles were analyzed in six tissues (i.e., flower, flower bud, leaf, stem, root 1, and root 2) in *I. triloba* and *I. trifida* based on RNA-seq data (Appendix A). In *I. triloba*, *ItbPIF1.2* and *ItbPIF3.2* presented a low expression level in all tissues, *IbPIF3.1* had a higher expression level in flower bud, and *ItbPIF4* was highly expressed in leaf (Appendix A). In *I. trifida*, *ItfPIF8* showed higher expression levels in all tissues, compared to other *ItfPIFs*, which was inconsistent with that in *I. triloba*. *ItfPIF1.1*, *ItfPIF3.1*, and *ItfPIF4* were highly expressed in leaf, whereas they showed low expression level in root. *ItfPIF1.2* was highly expressed in stem (Appendix A). These results suggested that *PIF* genes play different roles in the sweet potato and its two diploid relatives.

To verify the RNA-seq results, qRT-PCR was conducted to measure the expression levels of *IbPIF* genes in six tissues (i.e., shoot, leaf, petiole, stem, fibrous root, and mature tuberous root). The qRT-PCR results showed that *IbPIF1.2*, *IbPIF3.1*, and *IbPIF3.2* had lower expression in leaf, while *IbPIF8* had higher expression in leaf, *IbPIF1.1* and *IbPIF3.1* had a gradually down-regulated trend during root development, and *IbPIF4* was highly expressed in stem (Figure 6C). These results were roughly consistent with the RNA-seq analysis.

#### 2.7.2. Expression Analysis under Hormone Treatment

The synthesis and metabolism of plant hormones are involved in all aspects of plant development. Therefore, it is essential to explore the expression pattern of *PIFs* under hormonal treatments. The expression of *ItbPIFs* and *ItfPIFs* were analyzed based on public RNA-seq data of *I. triloba* and *I. trifida* under ABA, GA3, and IAA treatments. In *I. triloba*, compared with hormone stress control, *ItbPIFs* were induced by at least one hormone (except *ItbPIF8*) (Appendix A). Under ABA treatment, *ItbPIF3.1* and *ItbPIF3.2* were induced, while *ItbPIF1.1* and *ItbPIF4* were repressed. Under GA3 treatment, *ItbPIF1.2* was up-regulated, and *ItbPIF1.1*, *ItbPIF3.1*, and *ItbPIF3.2* were down-regulated. Under IAA treatment, *ItbPIF3.1* and *ItbPIF4* were up-regulated (Appendix A). In *I. trifida*, *ItfPIF1.1*, *ItfPIF1.2*, *ItfPIF3.1*, *ItfPIF3.2*, and *ItfPIF4* showed different expression patterns under hormone treatment, compared with those in *I. triloba*. *ItfPIF1.1* was up-regulated by ABA, while being repressed by GA3 and IAA. *ItbPIF1.2* was induced by ABA. *ItfPIF3.1* was up-regulated by GA3 and repressed by IAA. *ItfPIF3.2* was repressed by ABA, GA3, and IAA. *ItfPIF4* was repressed by ABA. *ItfPIF8* was repressed by ABA, GA3, and IAA (Appendix A). Overall, the expression patterns of *PIFs* in the two diploid relatives in response to ABA, GA3, and IAA are different, suggesting that *PIFs* are involved in different hormonal pathways between *I. triloba* and *I. trifida*.

We next investigated the relative expression of *IbPIFs* after 0 h, 0.5 h, 1 h, 3 h, 6 h, and 12 h of the hormone treatments by qRT-PCR, involving ABA, GA, IAA, MeJA, and SA (Figure 7). Under ABA treatment, all *IbPIFs* were down-regulated (Figure 7A). Under GA treatment, the majority of *IbPIFs* were significantly induced without *IbPIF1.2*, and the maximum values appeared at 0.5 h (Figure 7B). Under IAA treatment, *IbPIF1.1*, *IbPIF1.1*, *IbPIF1.2*, *IbPIF4*, and *IbPIF8* were significantly up-regulated, while *IbPIF3.1* was repressed (Figure 7C). Under MeJA treatment, *IbPIF1.1*, *IbPIF3.1*, *IbPIF3.2*, and *IbPIF8* were significantly up-regulated, while *IbPIF1.2* and *IbPIF4* were repressed (Figure 7D). When treated with SA, the expression levels of *IbPIFs* were significantly up-regulated. Among them, the expression levels of *IbPIF1.1*, *IbPIF1.2*, *IbPIF3.1*, and *IbPIF3.2* reached the peak at 12 h, while the expression levels of *IbPIF4* and *IbPIF8* reached the peak at 1 h. It is worth noting that the relative expression levels of *IbPIF1.1* and *IbPIF3.1* increased by 23.62-fold and 28.5-fold under SA treatment, respectively (Figure 7E). Together, six *IbPIF* genes were found to respond to two or more hormones, indicating that *IbPIFs* may participate in the cross-talk between various hormones.

#### 2.7.3. Expression Analysis under Abiotic Stresses

To explore the potential function of *PIFs* in responding to abiotic stress, we analyzed the expression patterns of *PIFs* using the RNA-seq data of *I. triloba* and *I. trifida* under salt, drought, cold, and heat stress treatments (Appendix A). Under cold and heat treatments, all of the *ItbPIF* genes were down-regulated compared with the control in *I. triloba* (Appendix A). In *I. triloba*, *ItbPIF3.1* was significantly up-regulated under salt and drought treatments. Under salt, drought, and cold treatments, the expression pattern of *ItfPIF1.1*, *ItfFIF1.2*, *ItfPIF3.1*, and *ItfPIF4* in *I. trifida* was the same as that of homologous genes in *I. triloba*, while the expression pattern of *ItfFIF3.2* and *ItbPIF3.2* was opposite (Appendix A). These results indicated that the *PIFs* in *I. triloba* and *I. trifida* present both commonalities and differences in response to abiotic stresses.

To further illustrate the effects of various abiotic stresses (i.e., NaCl, PEG, H_2_O_2_, cold, and heat) on the expression of *PIF* genes, we examined *IbPIFs* expression levels by qRT-PCR (Figure 8). Under salt stress treatment, all of the *IbPIFs* were up-regulated, with *IbPIF1.1*, *IbPIF3.1*, and *IbPIF8* peaking at 1 h, with *IbPIF1.2* and *IbPIF3.2* peaking at 12 h, with *IbPIF4* peaking at 0.5 h (Figure 8A). Under PEG stress treatment, *IbPIF1.1*, *IbPIF1.2*, *IbPIF3.1*, and *IbPIF3.2* were up-regulated and peaked at 3 h; in particular, *IbPIF3.1* was induced by more than 11.08-fold, while *IbPIF4* was repressed (Figure 8B). Under H_2_O_2_ stress treatment, all *IbPIFs* were significantly up-regulated, with *IbPIF1.2* especially induced by 12.98-fold (Figure 8C). Under cold stress treatment, more than half of the *IbPIFs* were induced, with *IbPIF3.2* up-regulated by 1.53-fold, while *IbPIF1.2* and *IbPIF8* were repressed (Figure 8D). Under heat stress treatment, *IbPIF3.1* was significantly up-regulated (by 5.64-fold) at 6 h, while *IbPIF1.1* and *IbPIF4* were up-regulated at 0.5 h (Figure 8E). In general, *IbPIF3.1* was induced by all five abiotic stress treatments in sweet potato, while *IbPIF8* was down-regulated under a majority of the abiotic stress treatments (PEG, H_2_O_2_, cold, and heat). These results indicated that *IbPIFs* might play a key role in abiotic stress resistance.

#### 2.7.4. Expression Analysis under Biotic Stresses

To understand the role of *PIF* genes under biotic stress, we analyzed the expression patterns of *PIFs* based on public RNA-seq data under beta-aminobutyric acid and benzothiadiazole S-methylester biotic stress treatments [57,58] (Appendix A). Under beta-aminobutyric acid biotic stress treatment, *ItbPIF3.2* was significantly induced, while *ItbPIF1.1*, *ItbPIF1.2*, *ItbPIF3.1*, and *ItbPIF4* were strongly suppressed. Under benzothiadiazole S-methylester biotic stress treatment, *ItbPIF1.2* and *ItbPIF3.1* were up-regulated, while *ItbPIF1.1*, *ItbPIF3.2*, and *ItbPIF4* were down-regulated (Appendix A). Under benzothiadiazole S-methylester biotic stress treatment, the expression of most *ItfPIF* genes was suppressed except for *ItfPIF1.2* and *ItfPIF8* in *I. trifida* (Appendix A).

To investigate the possible functions of *IbPIFs* under biotic stress, expression profiling of *PIF* genes was further analyzed in response to two common sweet potato diseases: *Fusarium* wilt disease and stem nematodes. The expression patterns of *IbPIF* genes at various time points were analyzed by qRT-PCR (Figure 9). After *Fob* infection, except for *IbPIF8*, which was down-regulated, the other five genes were up-regulated at all four time points, whereas *IbPIF1.1*, *IbPIF3.1*, *IbPIF3.2*, and *IbPIF4* were shown to be highly expressed and peaked at 0.5 d, after which they remained at a high level. The expression level of *IbPIF1.2* was slightly up-regulated, but decreased more significantly later under *Fob* infection (Figure 9A). After stem nematode infection, the expression levels of *IbPIF1.1*, *IbPIF1.2*, *IbPIF3.1*, and *IbPIF4* was significantly up-regulated and peaked at 4 d (Figure 9B). Meanwhile, the expression level of *IbPIF3.2* was up-regulated and peaked at 2 d. The *IbPIF3.1* was significantly up-regulated under abiotic and biotic stresses, so we selected *IbPIF3.1* for further research.

### 2.8. Overexpression of IbPIF3.1 Enhanced Drought Tolerance of Tobacco

To verify the effects of *IbPIF3.1* on abiotic stress, we obtained two transgenic lines of tobacco cv. Wisconsin 38 (W38) with overexpressed *IbPIF3.1* (OE1 and OE2). Under normal growth conditions, there were no significant differences between WT and transgenic plants in terms of root length and fresh weight (Figure 10A). Under PEG treatment, WT displayed severe growth retardation, whereas *IbPIF3.1*-OE lines formed new roots, where the length of the roots (Figure 10B) and fresh weights (Figure 10C) were significantly higher than that of WT.

In order to assess the degree of cell damage, we measured the malondialdehyde (MDA) and proline content. Under PEG treatment, the MDA content was significantly higher in WT than *IbPIF3.1*-OE lines (Figure 10D), while the proline content in WT was significantly lower than in *IbPIF3.1*-OE lines (Figure 10E). Further analysis showed that the overexpression of *IbPIF3.1* up-regulated the expression of stress-related genes *NtPOD*, *NtDREB1A*, *NtDREB1B*, and *NtDREB1D* under PEG treatment in transgenic plants (Figure 10F–I). These results suggested that overexpression of *IbPIF3.1* enhanced drought tolerance of tobacco.

### 2.9. Overexpression of IbPIF3.1 Enhanced Fob Resistance of Tobacco

In order to further analyze the function of *IbPIF3.1* in response to biotic stress, the *IbPIF3.1*-OE lines were infected with *Fusarium* wilt fungus. Before infection, there was no significant difference in morphology between WT and transgenic lines (Figure 11A,B). Eleven days after *Fob* infection, the leaves and stems of WT were withered and brown (Figure 11C). However, the withering degree of leaves and stems, and the number of diseased leaves of transgenic line OE2 were significantly lower than those of wild type, and maintained a good growth state (Figure 11D).

Further analysis showed that the overexpression of *IbPIF3.1* up-regulated the expression salicylic-acid responsive gene *NtPR1a* and hyper-sensitive response relative genes *NtHSR201* and *NtHSR515* after *Fob* infection in transgenic plants (Figure 11E–G). Overall, the results suggested that *IbPIF3.1* functions as a positive regulator of *Fob* resistance in tobacco.

## 3. Discussion

### 3.1. Identification and Evolution of PIFs Family

In this study, *PIFs* were identified in the cultivated hexaploid sweet potato and its two diploid relatives. The number of *PIF* genes in *I. batatas* was the same as that in its diploid relatives *I. triloba* (6) and *I. trifida* (6) (Figure 1, Appendix A). It is well known that cultivated sweet potato (*I. batatas*) originated from a hybrid between diploid and tetraploid ancestors, followed by a whole-genome duplication event. This process dates back to about 0.8 and 0.5 million years ago [52]. The number of *PIF* genes in sweet potato differs from that in other plants, such as *Arabidopsis* (8), rice (6), tea (7), tomato (8), grape (4), apple (7), potato (7), and carrot (5). According to the gene dosage balance hypothesis, genes in the same family are often functionally redundant [59]. Therefore, these quantitative differences might not affect the function of *PIFs* in plants. In this study, the results of phylogenetic analysis showed that IbPIFs had the closest relationship with tomato and potato PIFs (Figure 2). It is well known that sweet potato belongs to the family Convolvulaceae [53], and tomato and potato belong to the family Solanaceae. According to botanical classification, Solanaceae and Convolvulaceae belong to the Solanales [60,61]. Compared with other plants (in this study), they might have a closer relationship. In the future, it is possible to discover homologous genes from these closely related plant genomes and provide some reference for functional analysis.

PIFs directly interact with phytochrome to regulate the light signaling pathway, where phyA and phyB can specifically bind the APA and APB domains, respectively. In this study, we combined conserved domain analysis, and predicted motifs to further identify PIF family members in sweet potato [13,19]. All IbPIFs contained bHLH and APB domains. However, only IbPIF1.1, IbPIF3.1, and IbPF3.2 contained the APA domain (Figure 3A). The loss of the APA domain might lead to a loss of phyA binding ability, affecting light signal transduction [13].

### 3.2. The Expression of PIF Genes Was Tissue-Specific in Sweet Potato

In rice, *OsPIL13* is highly expressed in the node portions of the stem nodes, and overexpression of *OsPIL13* promotes internode elongation and reduces plant height [28]. *ZmPIF1* and *ZmPIF3* are highly expressed in pistils and leaves, and enhance grain yield by increasing the number of tillers and panicles [30,31,62]. *SlPIF4* is highly expressed in tomato leaves and fruits and decreases significantly after ripening [63]. In this study, the expression level of *PIFs* were found to be higher in the stem and leaf (especially in mature leaf) of sweet potato and its two diploid relatives *I. triloba* and *I. trifida*, and low in root (Figure 6, Appendix A). This is similar to the expression of *DcPIFs*, *MdPIFs*, and *CaPIF* genes [19,21,25]. Interestingly, the expression levels of *IbPIF 1.1* and *IbPIF3.1* were higher in the fibrous roots than in the tuberous roots in sweet potato (Figure 6). The difference in the expression of these genes might be related to the growth and development of sweet potato.

### 3.3. PIFs Play Important Roles in Hormone Signaling Pathways in Sweet Potato

In *Arabidopsis*, PIF3, RGA, and COI1 form a signal cascade to regulate plant defense and growth by interfering with JA and GA signals [64]. PIF can cooperate with JA and ethylene (ET) signaling to regulate *Arabidopsis* resistance to *Beauveria bassiana* [37]. In this study, we in silico predicted that IbPIFs could interact with hormone synthesis and signal transduction-related proteins, such as the GA signaling-related proteins RGA, RGL1, RGL2, and RGL3, and BR signaling pathway-related protein BZR1 (Figure 5). The complex interactions between PIFs and hormone signaling pathway proteins suggested that they might play an important role in regulating plant growth, development, and stress response.

PIFs are involved in multiple hormones signaling pathways [65], for example, AtPIF1 controls seed germination by regulating ABA and GA signaling pathways [66,67]. AtPIF4 and AtPIF5 can integrate light signals and auxin signaling pathways to regulate plant rhythmic growth [68]. In this study, we found that most *PIF* genes were induced by at least one hormone, and the promoter region of *PIFs* contained at least one hormone corresponding *cis*-element (Figure 4, Figure 7, Appendix A). In sweet potato, *IbPIF1.1*, *IbPIF3.2*, and *IbPIF8* were induced by GA, IAA, JA, and SA treatments (Figure 7). *IbPIF3.1* and *IbPIF4* were induced by GA, SA, and JA or IAA treatments (Figure 7). *IbPIF1.2* was induced by IAA and SA treatments (Figure 7). In addition, we found that some of the sweet potato diploid relatives’ homologous *PIF* genes showed different expression patterns in response to ABA, GA, and IAA treatments. Under ABA treatment, *ItbPIF1.2*, *ItfPIF1.2*, *ItbPIF3.1*, and *ItfPIF3.1* were up-regulated, while *IbPIFs* were not sensitive to ABA. Under GA or IAA treatment, *IbPIF1.1* showed opposite expression trends compared with *ItbPIF1.1*, and *ItfPIF1.1*. Under GA or IAA treatment, the expression trend of *ItbPIF3.1* was opposite to that of *ItfPIF3.1* (Figure 7, Appendix A). These results suggest that PIFs might be involved in the regulation network of different hormones in sweet potato and its wild relatives, thus affecting the growth and defense of plants.

### 3.4. PIF Genes Response to Multiple Stresses

*OsPIL14* is significantly induced by salt stress; the overexpression of *OsPIL14* enhances seedling growth under salt stress [69]. *DcPIF3*, *ZmPIF1*, and *ZmPIF3* were induced by drought treatment, significantly enhancing the drought tolerance of transgenic plants [25,30,31]. *AtPIF4* and *AtPIF7* were up-regulated under heat and cold treatment, which negatively regulated the freezing resistance of *Arabidopsis thaliana* [32,70]. *AtPIF1*, *AtPIF3*, *AtPIF4*, and *AtPIF5* were down-regulated after *Botrytis* inoculation, which negatively regulated the plant defense against *Botrytis cinerea* [37]. In this study, most *PIFs* were induced by salt, PEG, H_2_O_2_, cold, and heat stresses (Figure 8). For example, *IbPIF4* was up-regulated 2.38-fold within 0.5 h under NaCl treatment (Figure 8A), *IbPIF3.1* was up-regulated 11.08-fold within 3 h under PEG treatment (Figure 8B), *IbPIF1.2* was up-regulated 12.98-fold within 3 h under H_2_O_2_ treatment (Figure 8C), *IbPIF3.2* was up-regulated 1.53-fold within 3 h under cold treatment (Figure 8D), and *IbPIF3.1* was up-regulated 5.64-fold within 6 h under heat treatment (Figure 8E). The two diploid relatives, *ItbPIF1.1*, *ItbPIF3.1*, *ItfPIF1.1,* and *ItfPIF3.1* were up-regulated under NaCl and drought stress treatments, and *ItfPIF3.2* was up-regulated under cold and heat stress treatments (Appendix A). These results showed that PIFs might be involved in responses to abiotic stress in sweet potato and its wild relatives. In addition, the expression levels of *IbPIF1.1*, *IbPIF1.2*, *IbPIF3.1*, *IbPIF3.2*, and *IbPIF4* were up-regulated and maintained at high level for several days in sweet potato after *Fob* and stem nematode infections (Figure 9). In the two diploid relatives, the expression of *ItbPIF3.1*, *ItbPIF3.2*, and *ItfPIF8* genes were up-regulated under beta-aminobutyric acid and benzothiadiazole S-methylester biotic stresses treatment (Appendix A).

Finally, we developed a preliminary understanding of the regulatory pathways in which *PIF* genes may be involved by RNA-seq and qRT-PCR. However, there were a few discrepancies between data obtained from the qRT-PCR and RNA-seq, since the respective homologous genes showed differential expression under stress in *I. batatas*, *I. triloba*, and *I. trifida*. These results may reflect the differences in responses to environment stress between cultivated and wild species [71,72]. Studies have found that some wild species are better able than cultivated species to resist biotic and abiotic stresses [73,74,75]. Therefore, the diploid relatives are valuable resources for the improvement of cultivated sweet potato [76].

### 3.5. Overexpressing IbPIF3.1 Significantly Enhanced Drought Tolerance and Fob Resistance of Tobacco

Some studies have reported that PIFs are closely associated with abiotic stress responses [77]. OsPIL15 interacts with OsHHO3 to directly activate *OsABI5* expression and negatively regulate stomatal opening [78]. *ZmPIF1* and *ZmPIF3* enhance the drought tolerance of rice by reducing transpiration and the leaf water loss rate [30,31]. AtPIF1, AtPIF4, and AtPIF5 specifically bind to the G-box of the *CBF* gene and negatively regulate plant cold tolerance [32,33]. A recent study has found that phytochromes are also involved in this process. On the one hand, phyB directly activates *CBF* genes expression; on the other hand, phyB inhibits the interaction between PIF1, PIF4, PIF5, and CBF, further enhancing the cold tolerance of plants [79]. High temperature inactivates phyB, leading to accumulation of PIF4, thus promoting plant cell elongation and early flowering [80]. The overexpression of *DcPIF3* can significantly improve drought and salt tolerance in *Arabidopsis* [25]. In this study, the expression of *IbPIF3.1* was significantly up-regulated by NaCl, PEG, H_2_O_2_, cold, heat, *Fob* and stem nematodes, and its overexpression could significantly improve the drought resistance and *Fusarium* wilt resistance of transgenic tobacco (Figure 10 and Figure 11).

Drought can cause plant cells to produce excess reactive oxygen species (ROS) [81]. Excessive ROS can aggravate membrane lipid peroxidation, causing damage to the cell membrane, resulting in a large number of secondary products such as MDA in plants [82]. Some studies have shown that plants can improve stress resistance by accumulating osmotic adjustment substances, such as proline and soluble sugar, to regulate osmotic balance, activate the ROS scavenging system, and protect membrane integrity [83,84]. Antioxidant enzymes, such as SOD, POD, CAT, and APX, are employed to scavenge excessive ROS accumulated under abiotic stress in plants [85]. Previous studies have found that AtDREB1A and AtDREB2A play an important role in the abiotic stress response [86]. The overexpression of *AtDREB2A* transgenic lines improved the drought tolerance of *Arabidopsis* [87]. The overexpression of soybean *GmDREB1* genes enhances drought and cold tolerance in transgenic wheat plants [88]. In this study, under drought stress, *IbPIF3.1*-OE lines had a lower MDA content and higher proline content than that of WT plants (Figure 10D,E). The qRT-PCR results showed that the expression of *NtPOD*, *NtDREB1A*, *NtDREB1B*, and *NtDREB1D* was significantly up-regulated in the *IbPIF3.1*-OE tobacco plants compared to WT under drought stress (Figure 10F–I). These results showed that overexpression of *IbPIF3.1* could increase drought tolerance of the transgenic tobacco by improving reactive oxygen species (ROS) scavenging ability and modulating the expression of drought-related genes. 

However, there has been little research on the molecular mechanisms of PIFs involved in biotic stress responses. In *Arabidopsis*, PIF4 positively regulates the temperature-induced suppression of defense responses to *Pto* DC3000 [36]. AtPIFs negatively regulate plant defenses against *B. cinerea* by directly repressing *ERF1* expression [37]. Pathogen-related (PR) proteins are considered as major regulators of the defense system to increase plant disease resistance [89]. In wheat, *TdPR1.2* gene expression was strongly induced by SA to inhibit the growth of bacteria and fungi [90]. Hypersensitivity reaction (HR) is an important defense mechanism of plants under biotic stress [91]. It was found that *HSR201* and *HSR515* in tobacco were closely related to hypersensitivity [92]. In this study, the expression of *PR1a*, *HSR201*, and *HSR515* was significantly up-regulated in transgenic tobacco plants overexpressing *IbPIF3.1* compared with WT after *Fob* infection (Figure 11E–G). These results suggested that the enhanced *Fob* resistance of the transgenic plants’ overexpressing *IbPIF3.1* may be due to the increased expression of stress-related genes. Together, all of results indicated that PIFs are key signal integrators in biotic and abiotic pathways to regulate plant growth [62]. In order to better cope with future environmental challenges, the potential functions of *PIFs* need to be continuously explored.

## 4. Materials and Methods

### 4.1. Identification of PIFs

To identify the *PIF* genes in sweet potato, all protein sequences were acquired from Sweetpotato Genomics Resource (http://sweetpotato.uga.edu/index.shtml; accessed on 15 September 2022) and Ipomoea Batatas Genome Browser (http://public-genomes-ngs.molgen.mpg.de/sweetpotato/; accessed on 15 September 2022). Then, we downloaded the protein sequence of *Arabidopsis* PIFs from the TAIR Arabidopsis database (https://www.arabidopsis.org/; accessed on 15 September 2022) and blasted the homologous sequence against the sweet potato genome database. All obtained sequences next used the SMART program (https://smart.embl.de/; accessed on 17 September 2022) and Conserved Domain Database (https://www.ncbi.nlm.nih.gov/Struture/cdd/wrpsb.cgi; accessed on 17 September 2022) to confirm the presence of conserved bHLH and APB domains.

### 4.2. Protein Properties Prediction and Chromosomal Distribution of PIFs

The physiological and biochemical properties of IbPIF proteins, including molecular weight, isoelectric point, unstable index, and hydrophilicity, were determined using the ExPASy (https://web.expasy.org/compute_pi/; accessed on 19 September 2022). The subcellular localization of IbPIFs was predicted using WoLF PSORT (https://www.genscript.com/wolf-psort.html; accessed on 19 September 2022).

The *IbPIFs*, *ItbPIFs*, and *ItfPIFs* were separately mapped to the *I. batatas*, *I. triloba*, and *I. trifida* chromosomes based on the chromosomal locations provided in the Sweetpotato Genomics Resource (http://sweetpotato.uga.edu/index.shtml; accessed on 22 September 2022). The visualization was created using the TBtools software [93].

### 4.3. Phylogenetic Analysis of PIFs

All PIFs amino acid sequences of *A. thaliana* (At), *C. sinensis* (Cs), *D. carota* (Dc), *I. batatas* (Ib), *I. triloba* (Itb), *I. trifida* (Itf), *M. domestica* (Md), *O. sativa* (Os), *S. lycopersicum* (Sl), *S. tuberosum* (St), and *V. vinifera* (Vv) were aligned using ClustalX with default settings. A ML phylogenetic tree of the PIFs was constructed by MEGA 11.0 with the bootstrap test of 1000 [94]. The phylogenetic tree was constructed using ChiPlot (https://www.chiplot.online/; accessed on 26 September 2022).

### 4.4. Motifs Identification and Conserved Domain Analysis of PIFs

The conserved motifs of PIFs were analyzed using MEME (https://meme-suite.org/meme/tools/meme; accessed on 20 September 2022), where the maximum number of motif parameters was set to 10. The conserved domains and exon–intron structure of PIFs were visualized using the TBtools software.

### 4.5. Protein Interaction Network of PIFs

The protein interaction network of PIFs was predicted by GeneMAINA (http://genemania.org/; accessed on 1 October 2022) and String (https://www.string-db.org/; medium confidence 0.400; accessed on 1 October 2022), based on *Arabidopsis* orthologous proteins. The protein–protein interaction (PPI) network and node network diagrams were constructed using the Cytoscape software 3.2 (Institute for Systems Biology, Seattle, WA, USA) [95].

### 4.6. The qRT-PCR Analysis of PIFs

The stem nematode-tolerant sweet potato cv. Lushu 3 was used for qRT-PCR analysis in this study. In vitro grown Lushu 3 plants were cultured on Murashige and Skoog (MS) medium at 27 ± 1 °C under a photoperiod consisting of 13 h of cool-white, fluorescent light at 54 μmol m^−2^ s^−1^ and 11 h of darkness. The in vitro plantlets were subsequently cultivated in a field at the campus of China Agricultural University, Beijing, China.

The shoots, leaves, petioles, stems, fibrous root, and mature tuberous root of 3-month-old field-grown Lushu 3 plants were used for expression analysis of various tissues. For the expression analysis of hormone and abiotic treatment, four-week-old in vitro-grown plants were transferred into half strength MS medium containing 100 μM ABA, 100 μM GA, 100 μM IAA, 100 μM MeJA, 100 μM SA, 200 mM NaCl, 20% PEG6000, 10 mM H_2_O_2_, 4 °C or 35 °C for treatment, and the leaves were sampled at 0, 0.5, 1, 3, 6, and 12 h after treatment. For expression analysis after *Fob* infection (detailed in Section 4.9), fresh leaves of Lushu 3 were sampled at 0 d, 0.5 d, 1 d, 2 d, and 3 d after inoculation with *Fob* [43]. For expression analysis after stem nematode infection, the roots of Lushu 3 were sampled at 0 h, 6 h, 1 d, 2 d, 4 d, 6 d, 8 d, and 10 d after 500 sweet potato stem nematode infection, respectively [41]. Three independent biological replicates were conducted, each with three plants.

Total RNA was extracted using the TRIzol method (Invitrogen, Carlsbad, CA, USA). The reaction mixture was composed of first-strand cDNA, primer mix, and SYBR Green Real-Time PCR Master Mix (TaKaRa Biotech Dalian, China; code: DRR037A) to a final volume of 20 μL. A sweet potato *actin* gene (GenBank AY905538, 20, 5, 21) was used as an internal control. The relative gene expression levels were quantified with the comparative C_T_ method. The specific primers used in qRT-PCR analysis are listed in Appendix A. The qRT-PCR was conducted using the SYBR detection protocol on 7500 Real-Time PCR instrument (Applied Biosystems, Foster City, CA, USA). The heat maps of the gene expression profiles were constructed using the TBtools software.

### 4.7. Transcriptome Analysis

The RNA-seq data of *ItbPIFs* and *ItfPIFs* in *I. triloba* and *I. trifida* were downloaded from the Sweetpotato Genomics Resource (http://sweetpotato.uga.edu/index.shtml; accessed on 10 December 2022) [53]. The RNA-seq data of *IbPIFs* in Yan252 and Xuzi3 were obtained from related research [96]. The expression levels of *PIFs* were calculated as fragments per kilobase of exon per million fragments mapped (FPKM). The heat maps were constructed using the TBtools software.

### 4.8. Production of Transgenic Tobacco Plants

The CDS of *IbPIF3.1* was amplified from Lushu 3 and inserted into pBI121 binary vector (Appendix A). The *35S-IbPIF3.1-NOS* expression cassette was excised from the *pBI121-IbPIF3.1* vector and integrated to the pCAMBIA3301 vector, in order to obtain the overexpression vector *pC3301-121-IbPIF3.1*. This recombinant plasmid was transferred into *N. tabacum* cv. W38 by *A. tumefaciens*-mediated transformation method, in order to generate T_0_ transgenic tobacco lines overexpressing *IbPIF3.1*. Transgenic lines were verified by PCR amplification and qRT-PCR analysis [43].

### 4.9. PEG Stress Treatment

In vitro-grown *IbPIF3.1*-OE lines and WT were cultured on MS medium with or without (control) 20% PEG6000 at 27 ± 1 °C under a photoperiod consisting of 13 h of cool-white, fluorescent light at 54 μmol m^−2^ s^−1^ and 11 h of darkness. After four weeks, the root length and fresh weight were measured. Meanwhile, their leaf formation was investigated, and proline and MDA contents were measured [42]. Three independent biological replicates were conducted, each with three plants. The expression of stress-related genes was analyzed by qRT-PCR by above-mentioned method. The specific primers used in qRT-PCR analysis are listed in Appendix A.

### 4.10. Fusarium Wilt Resistance Assay

The *Fob* fungal culture was homogenized on a PDA plate, suspended in sterilized water, and adjusted to a spore density of 1.5 × 10^7^ mL^−1^. Then, 40-day-old *IbPIF3.1*-OE lines and WT cuttings (without roots) were immersed into the spore solution for 30 min and incubated into sterilized sand moistened with sterilized Hoagland solution [43]. After 11 days, the number of diseased leaves was recorded. Three independent biological replicates were conducted, each with three plants. The expression of stress-related genes was analyzed by qRT-PCR by above-mentioned method. The specific primers used in qRT-PCR analysis are listed in Appendix A.

## 5. Conclusions

In this study, *PIF* genes from sweet potato and its two diploid relatives were identified, and their protein properties, chromosomal location, phylogenetic relationships, gene structure, promoter *cis*-elements, protein interaction network, and expression patterns were comprehensively and systematically investigated. Our research revealed that *IbPIFs* participate in plant growth and development, as well as responses to abiotic and biotic stresses. The functional analysis results indicated that the overexpression of *IbPIF3.1* significantly improves the drought tolerance and *Fusarium* wilt resistance of transgenic tobacco plants. This study lays a foundation for further analysis of the function of *PIF* genes and is of great significance for the stress-resistant genetic engineering of sweet potato.

## Figures and Tables

**Figure 1 ijms-24-04092-f001:**
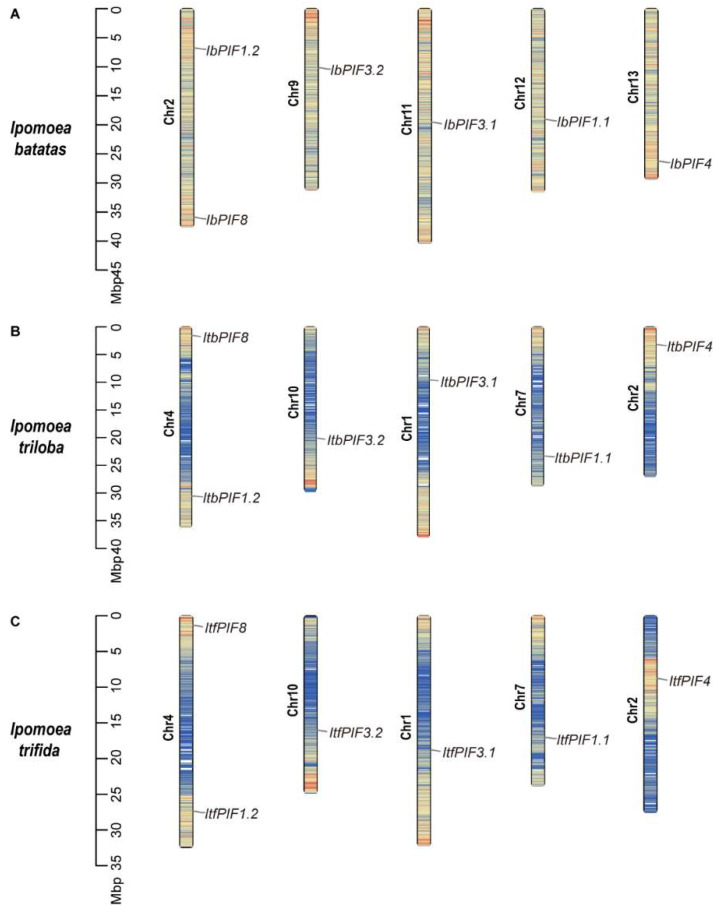
Chromosomal location and distribution of *PIF* genes in (**A**) *I. batatas*; (**B**) *I. triloba*; and (**C**) *I. trifida*. The bars represent chromosomes, the chromosome numbers are displayed on the left side, and the gene names are displayed on the right side. The relative chromosomal location of each *PIF* gene is marked with the black line at the right side.

**Figure 2 ijms-24-04092-f002:**
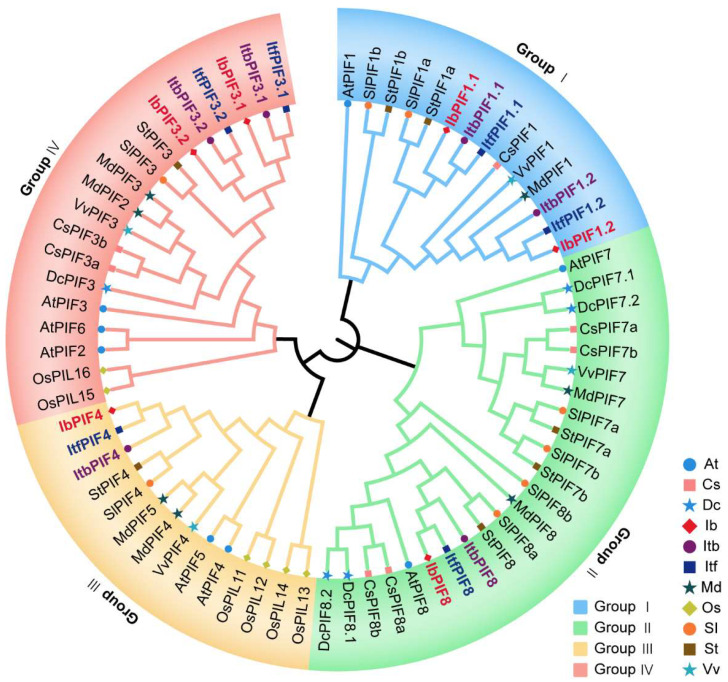
Phylogenetic analysis of the PIF families in *A. thaliana*, *C. sinensis*, *D. carota*, *I. batatas*, *I. triloba*, *I. trifida*, *M. domestica*, *O. sativa*, *S. lycopersicum*, *S. tuberosum*, and *V. vinifera*. A total of 70 PIFs were divided into four groups (groups I–IV), according to the evolutionary distance. The blue circles represent the eight AtPIFs in *A. thaliana*. The pink squares represent the seven CsPIFs in *C. sinensis*. The blue stars represent the five DcPIFs in *D. carota*. The red rhombus represents the six IbPIFs in *I. batatas*. The purple circles represent the six ItbPIFs in *I. triloba*. The indigotin squares represent the six ItfPIFs in *I. trifida*. The dark green stars represent the seven MdPIFs in *M. domestica*. The yellow rhombus represents the six OsPIFs in *O. sativa*. The orange circles represent the eight SlPIFs in *S. lycopersicum*. The brown squares represent the seven StPIFs in *S. tuberosum*. The light green stars represent the four VvPIFs in *V. vinifera*.

**Figure 3 ijms-24-04092-f003:**
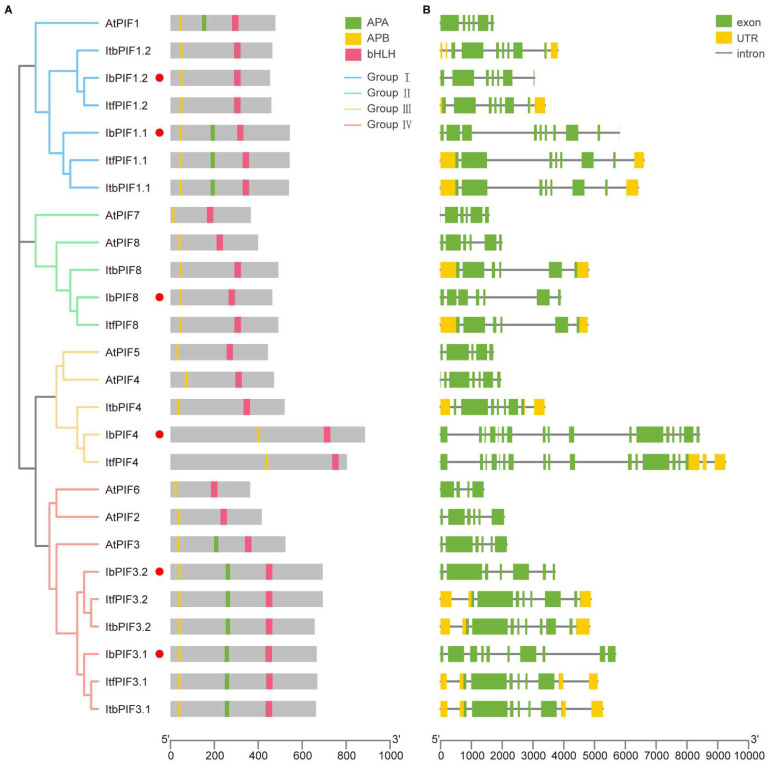
The phylogenetic tree showing that PIFs are distributed into four groups on the left. The red circle represents the IbPIFs. (**A**) Conserved domain structure of PIFs in *A. thaliana*, *I. batatas*, *I. triloba*, and *I. trifida*. The yellow, green, and pink boxes represent the APB domain, APA domain, and bHLH domain, respectively; and (**B**) exon–intron structure of *PIFs* in *A. thaliana*, *I. batatas*, *I. triloba*, and *I. trifida*. The yellow boxes, green boxes, and grey lines represent the UTR, exons, and introns, respectively.

**Figure 4 ijms-24-04092-f004:**
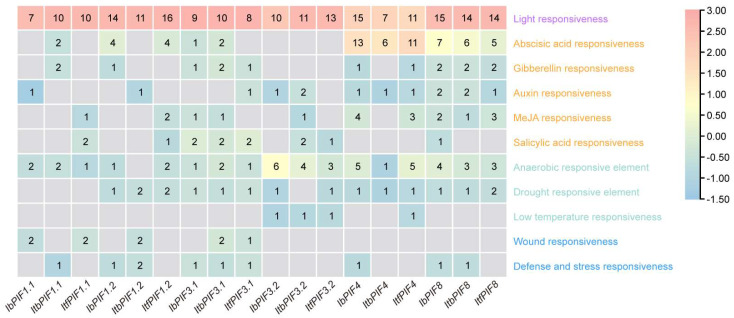
Analysis of *cis*-elements of *IbPIFs* in *I. batatas*, *I. triloba,* and *I. trifida*. The degree of red color represents the number of *cis*-elements upstream of the *PIFs*.

**Figure 5 ijms-24-04092-f005:**
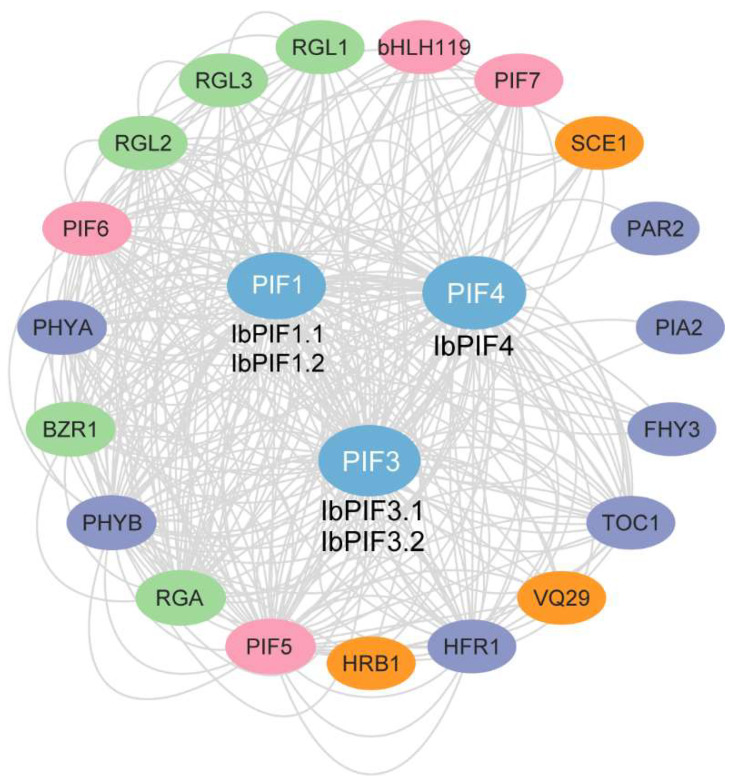
Functional interaction networks of IbPIFs in *I. batatas* according to orthologues in *A. thaliana.* Network nodes represent proteins, and lines represent protein–protein associations.

**Figure 6 ijms-24-04092-f006:**
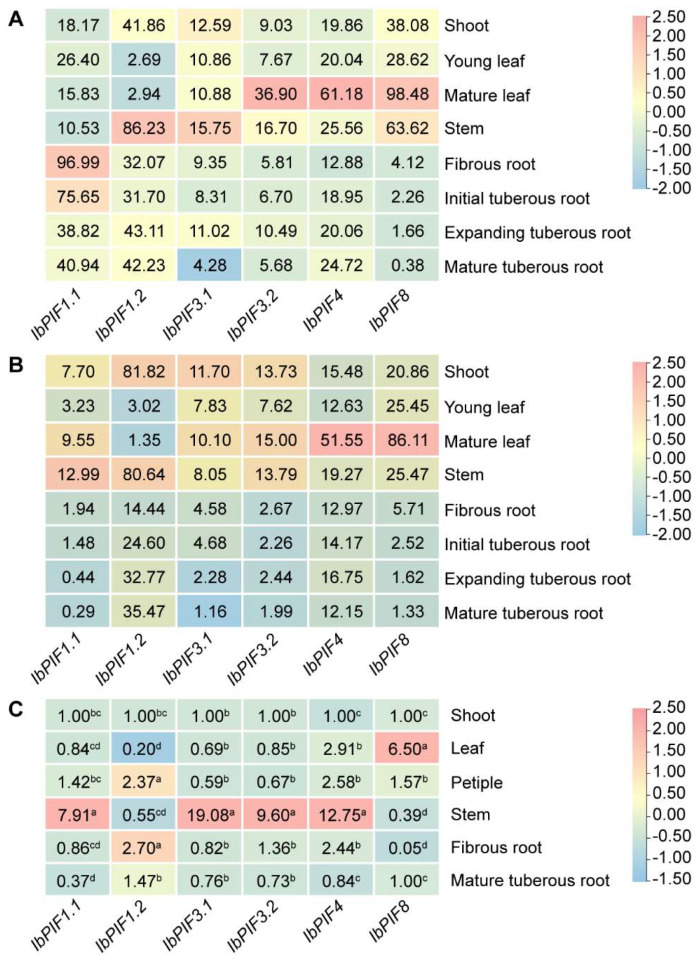
Gene expression patterns of *IbPIFs* in different tissues (shoot, young leaf, mature leaf, stem, fibrous root, initial tuberous root, expanding tuberous root, and mature tuberous root) of (**A**) Yan252 and (**B**) Xuzi3, as determined by RNA-seq. Log_2_(FPKM + 1) is shown in the boxes. (**C**) Gene expression patterns of *IbPIFs* in shoot, petiole, leaf, stem, fibrous root, and mature tuberous root of *I. batatas.* The values were determined by qRT-PCR from three biological replicates consisting of pools of three plants, and the results were analyzed using the comparative C_T_ method. The expression at 0 h in each treatment was considered “1”. The fold change is shown in the boxes. Different lowercase letters indicate a significant difference of each *IbPIFs* at *p* < 0.05 based on Student’s *t*-test.

**Figure 7 ijms-24-04092-f007:**
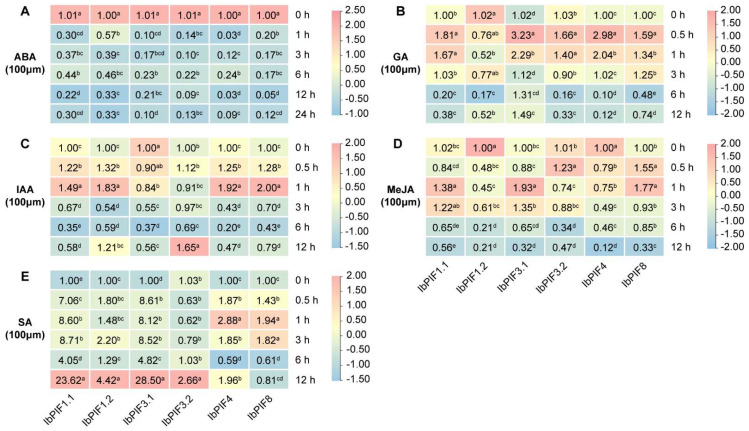
Gene expression patterns of *IbPIFs* in response to different phytohormones in *I. batatas*: (**A**) ABA; (**B**) GA; (**C**) IAA; (**D**) MeJA; and (**E**) SA. The values were determined by qRT-PCR from three biological replicates consisting of pools of three plants, and the results were analyzed using the comparative C_T_ method. The expression at 0 h in each treatment was considered “1”. The fold change is shown in the boxes. Different lowercase letters indicate a significant difference of each *IbPIFs* at *p* < 0.05 based on Student’s *t*-test.

**Figure 8 ijms-24-04092-f008:**
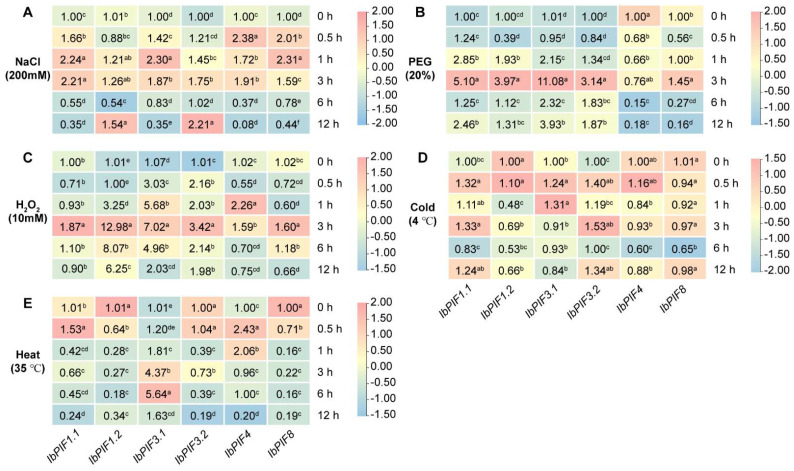
Gene expression patterns of *IbPIFs* of *I. batatas* in response to abiotic stresses: (**A**) NaCl; (**B**) PEG; (**C**) H_2_O_2_; (**D**) cold; and (**E**) heat. The values were determined by qRT-PCR from three biological replicates consisting of pools of three plants, and the results were analyzed using the comparative C_T_ method. The expression at 0 h in each treatment was considered “1”. The fold change is shown in the boxes. Different lowercase letters indicate a significant difference of each *IbPIFs* at *p* < 0.05 based on Student’s *t*-test.

**Figure 9 ijms-24-04092-f009:**
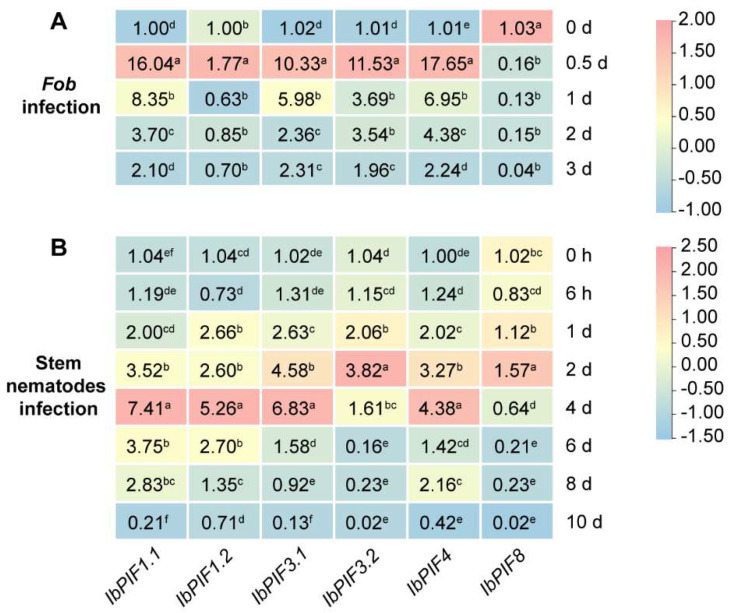
Gene expression patterns of *IbPIFs* of *I. batatas* in response to biotic stresses: (**A**) *Fob* infection; and (**B**) stem nematode infection. The values were determined by qRT-PCR from three biological replicates consisting of pools of three plants, and the results were analyzed using the comparative C_T_ method. The expression at 0 h in each treatment was considered “1”. The fold change is shown in the boxes. Different lowercase letters indicate a significant difference of each *IbPIFs* at *p* < 0.05 based on Student’s *t*–test.

**Figure 10 ijms-24-04092-f010:**
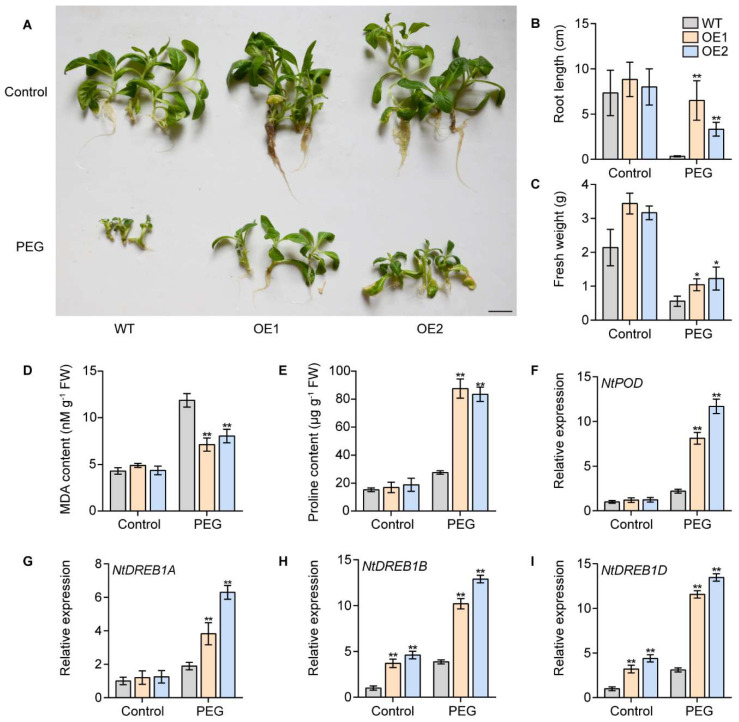
Drought tolerance identification of WT and *IbPIF3.1*-OE transgenic tobacco plants cultured on MS medium without (control) or with 20% PEG6000 for 4 weeks: (**A**) phenotypes; (**B**) root length; (**C**) fresh weight (scale bar = 2.5 cm); (**D**) MDA content; and (**E**) proline content in the leaves of plants after 4 weeks of treatment. Transcript levels of (**F**) *NtPOD*; (**G**) *NtDREB1A*; (**H**) *NtDREB1B*; and (**I**) *NtDREB1D* in the leaves of plants after 4 weeks of treatment. The transcript levels of the genes in WT without treatment control were set to 1. The values were determined by qRT-PCR from three biological replicates consisting of pools of three leaves. The error bars indicate ± SD (*n* = 3). *, *p* < 0.05; **, *p* < 0.01; Student’s *t*–test.

**Figure 11 ijms-24-04092-f011:**
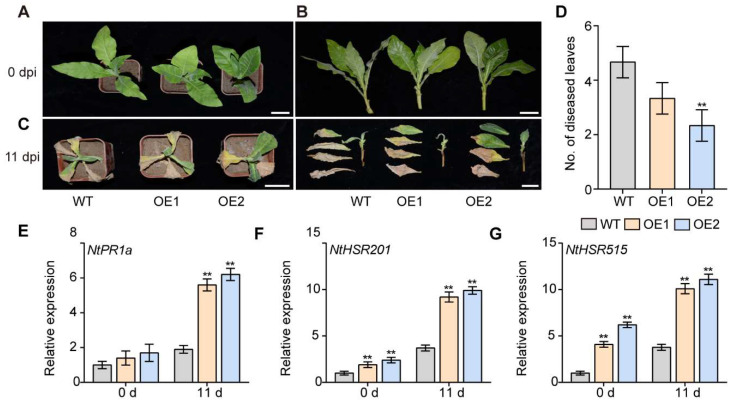
Development of plant disease symptoms in WT and *IbPIF3.2*-OE transgenic tobacco plants after *Fob* inoculation. W38 and *IbPIF3.1*-OE transgenic plants were inoculated with *Fob* spores at a density of 1.5 × 10^7^ mL^−1^ for 11 d: (**A**,**B**) phenotypes; (**C**) development of disease symptoms in leaves of WT and *IbPIF3.1*-OE transgenic tobacco lines after *Fob* inoculation (scale bar = 2.5 cm); (**D**) the number of diseased leaves in WT and *IbPIF3.1*-OE transgenic tobacco lines at 11 d; and transcript levels of (**E**) *NtPR1a*; (**F**) *NtHSR201*; and (**G**) *NtHSR515* in WT and *IbPIF3.1*-OE transgenic tobacco lines. The transcript levels of genes in WT before inoculation were set to 1. The values were determined by qRT-PCR from three biological replicates consisting of pools of three leaves. The error bars indicate ± SD (*n* = 3). **, *p* < 0.01; Student’s *t*–test.

**Table 1 ijms-24-04092-t001:** Characterization of IbPIFs in sweet potato.

Gene ID	Gene	CDS	Protein	Genomic	MW	pI	Instability	Gravy	Subcellular	Best Hits	*Arabidopsis*
	Name	(bp)	(aa)	(bp)	(kDa)						Gene ID
Ib12g49455	*IbPIF1.1*	1635	544	8025	59.534	5.79	62.27	−0.583	nucleus	*AtPIF1*	At2g20180
Ib02g5235	*IbPIF1.2*	1365	454	3275	48.728	5.29	55.99	−0.508	nucleus	*AtPIF1*	At2g20180
Ib11g44153	*IbPIF3.1*	2097	698	6758	663	5.98	54.32	−0.58	nucleus	*AtPIF3*	At1g09530
Ib09g35474	*IbPIF3.2*	2085	694	4825	657	7.17	56.32	−0.581	nucleus	*AtPIF3*	At1g09530
Ib13g54841	*IbPIF4*	2661	886	8671	97.031	6.58	43.27	−0.518	nucleus	*AtPIF4*	At2g43010
Ib02g9143	*IbPIF8*	1395	464	4500	49.17	6.55	54.59	−0.545	nucleus	*AtPIF8*	At4g00050

CDS, coding sequence; MW, molecular weight; pI, isoelectric point.

## Data Availability

The data presented in this study are available on request from the corresponding author.

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
