# Peer review of "Genome-Wide Characterization of the PIFs Family in Sweet Potato and Functional Identification of IbPIF3.1 under Drought and Fusarium Wilt Stresses"

_ijms, 2023, doi:10.3390/ijms24044092_

Round 1
Reviewer 1 Report
Two minor comments:
Firstly, PIFs from sweet potato has a closer relationship with tomato and potato is consistent with their taxonomic relationship. Solanaceae and Convolvulaceae are both under Solanales. This point should be briefly elaborated in discussion.
Secondly, it’s known that PIF affects plant growth through hormonal pathways, through which plants can be regulated in adaptation to biotic and abiotic stresses. In the present study, was there any other detectable phenotypic changes (in addition to resistances) in the transgenic tobacco? Please briefly discuss how future studies should be planned to better understand the molecular mechanism of the PIF-mediated resistance in sweetpotato.
Author Response
Response to Reviewer 1 Comments
Reviewer 1:
Two minor comments:
Firstly, PIFs from sweet potato has a closer relationship with tomato and potato is consistent with their taxonomic relationship. Solanaceae and Convolvulaceae are both under Solanales. This point should be briefly elaborated in discussion.
Response:
Thank you very much for your comments. Based on the comments, we have made careful modifications in the discussion as follow:
“In this study, the results of phylogenetic analysis showed that IbPIFs had a closer relationship with tomato and potato (Figure 2). It is well known that sweet potato belongs to the family Convolvulaceae [53], and tomato and potato belong to the family Solanaceae. According to botanical classification, Solanaceae and Convolvulaceae belong to the Solanales [58,59]. Compared with other plants (in this study), they might have a closer relationship. In the future, it is possible to discover homologous genes from these close plant genomes and provide some reference for functional analysis.”
(Please see lines 465-472)
Secondly, it’s known that PIF affects plant growth through hormonal pathways, through which plants can be regulated in adaptation to biotic and abiotic stresses. In the present study, was there any other detectable phenotypic changes (in addition to resistances) in the transgenic tobacco? Please briefly discuss how future studies should be planned to better understand the molecular mechanism of the PIF-mediated resistance in sweetpotato.
Response:
Thank you very much for your comments. PIFs may affect plant growth through hormonal pathways. Through hormonal pathways, plants can change their physiological and morphological characteristics to adapt to adversity (Zhu et al, 2002). Some studies have found that the hypocotyl lengths of AtPIF3-OX Arabidopsis plants were longer than WT grown under white light (Kim et al, 2003). In rice, the stem lengths of the OsPIL1-OXs transgenic line were significantly longer than the control plant after 30 d growth (Todaka et al, 2012). A previous study found that some transgenic plants show no obvious morphological differences compared with WT under normal growth conditions. Transgenic experiments indicated that the overexpression of ZmPIF3 in rice enhanced tolerance to drought and salt stresses without growth retardation compared with WT (Gao et al, 2015). In this study, we found that the phenotype of IbPIF3.1-OE transgenic tobacco plants was not significantly different from that of WT.
In the future, we plan to transfer IbPIF3.1 gene into sweet potato, and obtain overexpression, RNAi plants or CRISPR/Cas9 editing transgenic plants. Further analysis of how this gene responds to different stresses in hormone-mediated signaling pathways. Next, we will combine transcriptome, proteome, metabonomic to verify the regulatory mechanism of this gene.
Based on the comments, we have made careful modifications in the revised manuscript as follow:
“Under normal growth conditions, there were no significant differences between WT and transgenic plants in terms of root length and fresh weight (Figure 10A).”
(Please see lines 406-407)
References:
- Zhu, J.K. Salt and drought stress signal transduction in plants. Annu. Rev. Plant Biol. 2002, 53, 247-273, doi:10.1146/annurev.arplant.53.091401.143329.
- Kim, J.Y.; Yi, H.K.; Choi, G.; Shin, B.; Song, P.S.; Choi, G.S. - Functional characterization of phytochrome interacting factor 3 in phytochrorne-mediated light signal transduction. 2003, - 15, - 2407.
- Todaka, D.; Nakashima, K.; Maruyama, K.; Kidokoro, S.; Osakabe, Y.; Ito, Y.; Matsukura, S.; Fujita, Y.; Yoshiwara, K.; Ohme-Takagi, M.; et al. Rice phytochrome-interacting factor-like protein OsPIL1 functions as a key regulator of internode elongation and induces a morphological response to drought stress. Proc. Natl. Acad. Sci. U. S. A. 2012, 109, 15947-15952, doi:10.1073/pnas.1207324109.
- Gao, Y.; Jiang, W.; Dai, Y.; Xiao, N.; Zhang, C.Q.; Li, H.; Lu, Y.; Wu, M.Q.; Tao, X.Y.; Deng, D.X.; et al. A maize phytochrome-interacting factor 3 improves drought and salt stress tolerance in rice. Plant Mol.Biol. 2015, 87, 413-428, doi:10.1007/s11103-015-0288-z.

Reviewer 2 Report
Dear Editor.
In the article, the Authors have done research on PIF genes in sweet potato. PIF genes are an important gene sub-family of bHLH and were linked to abiotic and biotic stress function in some plants. The authors have comprehensively investigated the functional and evolutional characteristics of these genes. I believe this article is relatively well-done but with few minor corrections to be made which I have listed below.
Line 120-121, the sentence is misleading, rewrite it in more clear manner.
Line 122-125, the sentence does not mention the numbering of the rest of the PIFs.
Line 155-164, the paragraph does not clearly mention the method grouping was done.
Figure2, Group names on the figure can be placed in more symmetrical way. In the figure description, rewrite the name of species and genes in italic form (this must be checked throughout the paper).
Figure3, the model used for constructing the phylogenetic tree is not identical to the one used in phylogenetic tree in Figure 2.
Line 226-230, please cite the reference for grouping of plant hormonal response elements.
Figure 5, the sentence can be rewritten in clearer format, the authors mentioned two types of lines, but only grey lines are shown.
Line 453-455, please explain “normal growth” in the sense of infection level compared to WT.
Line 721. Section 4.8, please mention at which generation of plants were the research done on. T0 or T1.
Author Response
Response to Reviewer 2 Comments
Reviewer 2:
In the article, the Authors have done research on PIF genes in sweet potato. PIF genes are an important gene sub-family of bHLH and were linked to abiotic and biotic stress function in some plants. The authors have comprehensively investigated the functional and evolutional characteristics of these genes. I believe this article is relatively well-done but with few minor corrections to be made which I have listed below.
- Line 120-121, the sentence is misleading, rewrite it in more clear manner.
Response:
Thank you very much for your comments. Based on the comments, we have made careful modifications in the revised manuscript as follow:
“A total of 6, 6, and 6 PIF genes were identified from the cultivated hexaploid sweet potato (I. batatas) and its two wild relatives, I. triloba, and I. trifida, respectively.”
(Please see lines 116-118)
- Line 122-125, the sentence does not mention the numbering of the rest of the PIFs.
Response:
Thank you very much for your comments. In this study, based on the sequence homology analysis and gene structure analysis of the PIF homologous genes from Arabidopsis, we found that the PIF genes from sweet potato and its wild relatives have a high matching degree with the PIF1, PIF3, PIF4 and PIF8 of Arabidopsis, but a low matching degree with the other PIFs of Arabidopsis, so we named the PIFs of sweet potato and its wild relatives PIF1.1, PIF1.2, PIF3.1, PIF3.2, PIF4, PIF8, respectively. Therefore, other PIF genes of Arabidopsis are not described in this part.
Based on the comments, we have made careful modifications in the revised manuscript as follow:
“Based on sequence homologous and gene structure analysis with Arabidopsis PIFs, these genes were named PIF1.1, PIF1.2, PIF3.1, PIF3.2, PIF4, and PIF8, respectively. The PIF genes from I. batatas were named after “Ib”; I. triloba, named after “Itb”; and I. trifida, named after “Itf” (Supplementary Table S1).”
(Please see lines 118-122)
- Figure2, Group names on the figure can be placed in more symmetrical way. In the figure description, rewrite the name of species and genes in italic form (this must be checked throughout the paper).
Response:
Thank you very much for your comments. Based on comments, we have made careful modifications in the revised manuscript. Please see Figure 2, lines 164-174, 639-642.
- Figure3, the model used for constructing the phylogenetic tree is not identical to the one used in phylogenetic tree in Figure 2.
Response:
Thank you very much for your comments. Based on comments, we have made careful modifications in the revised manuscript. Please see Figure 3.
- Line 226-230, please cite the reference for grouping of plant hormonal response elements.
Response:
Thank you very much for your comments. Based on comments, we have made careful modifications in the revised manuscript as follow: Please see line 218.
- Figure 5, the sentence can be rewritten in clearer format, the authors mentioned two types of lines, but only grey lines are shown.
Response:
Thank you very much for your comments. Based on the comments, we have deleted the sentence "The gray lines represent physical interactions." in order to clearly express the content of Figure 5.
- Line 453-455, please explain “normal growth” in the sense of infection level compared to WT.
Response:
Thank you very much for your comments. In this study, after Fob infection for 11 days, the leaves and stems of WT withered and turned brown (Figure. 11C). The degree of withering on the leaves and stems, and the number of diseased leaves of transgenic line (OE2) were significantly lower than that of the WT, and maintained a good growth state.
Based on the comments, we have made careful modifications in the revised manuscript as follow:
“Eleven days after Fob infection, the leaves and stems of WT were withered and brown (Figure 11C). However, the withering degree of leaves and stems, and the number of diseased leaves of transgenic line OE2 were significantly lower than those of wild type, and maintained a good growth state (Figure 11D).”
(Please see lines 433-436)
- Line 721. Section 4.8, please mention at which generation of plants were the research done on. T0 or T1.
Response:
Thank you very much for your suggestions. In this study, we obtained T0 transgenic tobacco plants and identified them for drought tolerance and disease resistance.
Based on the comments, we have made careful modifications in the revised manuscript. Please see line 699.

Reviewer 3 Report
This paper provide new data on the genome-wide characterization in 3 different Ipomoea species of the PIFs gene familly, associated with very complete gene expression profiling under developmental and both biotic and abiotic stress conditions. After that, a functional characterization of the IbPIF3.1 from Ipomoea batatas was done in overexpressing transgenic plants, supporting the involvement of this gene in the response to various stress.
The paper is well written, easy to read and the results presented are complete and convincing.
I have only some minor changes to suggest:
Line 47: PHY (phy is written)
Line 88: delete ", and so on", "including" is already written
Table 1: include A. thaliana gene ID number (in the form of AtXgYYY)
Figure 2 legend: plant species names in italic (check in the text as well)
Please define "mature leaf" - is the leave are senescent or still photosynthetic or other?
Author Response
Response to Reviewer 3 Comments
Reviewer 3:
This paper provide new data on the genome-wide characterization in 3 different Ipomoea species of the PIFs gene familly, associated with very complete gene expression profiling under developmental and both biotic and abiotic stress conditions. After that, a functional characterization of the IbPIF3.1 from Ipomoea batatas was done in overexpressing transgenic plants, supporting the involvement of this gene in the response to various stress.
The paper is well written, easy to read and the results presented are complete and convincing.
I have only some minor changes to suggest:
- Line 47: PHY (phy is written)
Response:
Thank you very much for your comments. Based on the comments, we have made careful modifications in the revised manuscript. Please see line 44.
- Line 88: delete ", and so on", "including" is already written
Response:
Thank you very much for your comments. Based on the comments, we have made careful modifications in the revised manuscript.
- Table 1: include A. thaliana gene ID number (in the form of AtXgYYY)
Response:
Thank you very much for your comments. Based on the comments, we have made careful modifications in the revised manuscript. Please see table 1.
- Figure 2 legend: plant species names in italic (check in the text as well)
Response:
Thank you very much for your comments. Based on the comments, we have made careful modifications in the revised manuscript. Please see lines 164-174, 639-642.
- Please define "mature leaf" - is the leave are senescent or still photosynthetic or other?
Response:
Thank you very much for your comments. In this manuscript, we downloaded transcriptome data based on the research of Sun et al (2022) to analyze the expression of PIF genes in different tissues of sweet potato, including mature leaves (but there is no description of mature leaves, which might be photosynthetic, in the research of Sun et al). According to the methods of Huang et al. (2021) and Li et al. (2022), we selected the first to third fully expanded leaves of sweet potato cv. Lushu 3 transplanted for 90 days for the expression analysis of IbPIFs. The above analysis results showed that the expression pattern of PIF genes in fully expanded leaves was not completely consistent with that in mature leaves. The expression differences of these PIF genes need further analysis and identification.
References:
- Huang, Z.W.; Wang, Z.; Li, X.; He, S.Z.; Liu, Q.C.; Zhai, H.; Zhao, N.; Gao, S.P.; Zhang, H. Genome-Wide Identification and Expression Analysis of JAZ Family Involved in Hormone and Abiotic Stress in Sweet Potato and Its Two Diploid Relatives. Int. J. Mol. Sci. 2021, 22, doi:10.3390/ijms22189786.
- Li, X.; Zhao, L.M.; Zhang, H.; Liu, Q.C.; Zhai, H.; Zhao, N.; Gao, S.P.; He, S.Z. Genome-Wide Identification and Characterization of CDPK Family Reveal Their Involvements in Growth and Development and Abiotic Stress in Sweet Potato and Its Two Diploid Relatives. Int. J. Mol. Sci. 2022, 23, doi:10.3390/ijms23063088.
- Sun, H.Y.; Mei, J.Z.; Zhao, W.W.; Hou, W.Q.; Zhang, Y.; Xu, T.; Wu, S.Y.; Zhang, L. Phylogenetic Analysis of the SQUAMOSA Promoter-Binding Protein-Like Genes in Four Ipomoea Species and Expression Profiling of the IbSPLs During Storage Root Development in Sweet Potato (Ipomoea batatas). Front. Plant Sci. 2022, 12, 19, doi:10.3389/fpls.2021.801061.

Round 2
Reviewer 2 Report
I would like to thank the authors for their careful revision. Now, the manuscript is suitable for publishing in IJMS.
Author Response
Response to Reviewer 2 Comments
Reviewer 2:
Comments and Suggestions for Authors
I would like to thank the authors for their careful revision. Now, the manuscript is suitable for publishing in IJMS.
Response:
Thank you for recognizing our paper. We sincerely value your input to help improve the paper content.
